Incidence, persistence, and clearance of cervical human papillomavirus infection among gynecological outpatients in Kunming, Yunnan, China, 2019–2023: a retrospective cohort study

Huang Yafei 1 2
Wei Xiangcong 3
Guo Yan 2 4
Su Ting 1 2
Duan Qiuting 1 2
Fan Xin 1 2
Wan Jinxiu 1 2
Zhang Yufan 1 2
Zhang Guiqian 1 2
Sun Yi 1 2
Xu Ya xuya20090821hit@126.com 1 2
1 Department of Clinical Laboratory, The First People’s Hospital of Yunnan Province , Kunming , Yunnan , China
2 The Affiliated Hospital of Kunming University of Science and Technology , Kunming , Yunnan , China
3 Medical School, Kunming University of Science and Technology , Kunming , Yunnan , China
4 Department of Gynaecology and Obstetrics, The First People’s Hospital of Yunnan Province , Kunming , Yunnan , China
Anson Lesley
Electronic publication date: 2025 Nov 4
Publication date: 2025
Volume: 13
Electronic Location ID: e20215
Received 2025 May 9; Accepted 2025 Sep 19
Copyright: ©2025 Huang et al.
Copyright year: 2025
Copyright holder: Huang et al.
License: This is an open access article distributed under the terms of the Creative Commons Attribution License, which permits unrestricted use, distribution, reproduction and adaptation in any medium and for any purpose provided that it is properly attributed. For attribution, the original author(s), title, publication source (PeerJ) and either DOI or URL of the article must be cited.
License URL: https://creativecommons.org/licenses/by/4.0/

Keywords: HPV, Incidence, Persistence, Clearance, Yunnan

Funding: Yunnan Provincial Key Laboratory of Clinical Virology 202002AG070062 Research Fund for the Doctoral Program of the First People Hospital of Yunnan Province KHBS-2020-009 The National Natural Science Foundation of China 82060664 This work was supported by Yunnan Provincial Key Laboratory of Clinical Virology (grant No. 202002AG070062), Research Fund for the Doctoral Program of the First People Hospital of Yunnan Province (grant No. KHBS-2020-009), and the National Natural Science Foundation of China (grant No. 82060664). The funders had no role in study design, data collection and analysis, decision to publish, or preparation of the manuscript.

==============================
Human papillomavirus (HPV), a leading sexually transmitted pathogen, is characterized by persistent infection, which represents a critical risk factor for cervical carcinogenesis. This retrospective cohort study investigated the epidemiology of HPV among 45,149 gynecological outpatients in Yunnan, China (2019–2023). The 12-month cumulative incidence of HPV infection was 36.84%, with the highest rates observed in the 30–49-year age group. HPV-52 was the predominant subtype, followed by HPV-51, -81, -58, and -16. Persistent infection was observed in 55.56% of cases, most frequently involving HPV-42, -52, -58, -81, and -56, with higher rates in individuals younger than 30 and older than 59 years. Overall clearance reached 74.43% and was inversely correlated with age. Rapid clearance was predominantly observed for HPV-26, -83, -11, -82, and -44, whereas high-risk HPV types (HPV-58, HPV-52, HPV-35) and low-risk types (HPV-42, HPV-81, HPV-43) exhibited prolonged persistence. Regional data indicate elevated risks of incident and persistent infections with HPV-58, HPV-52, HPV-42, and HPV-81, with older populations showing greater susceptibility to persistence and younger individuals demonstrating faster clearance. These findings underscore the age-specific dynamics of HPV infection and highlight priority subtypes for regional cervical cancer prevention strategies.

Introduction

Human papillomavirus (HPV) infection is one of the most prevalent sexually transmitted infections (STIs) worldwide. Accumulated evidence to date indicates that HPV infection plays a central role in the development of cervical cancer (Jain et al., 2023). Despite global efforts to promote prophylactic vaccination, the uptake remains low, particularly in underdeveloped countries. Moreover, effective antiviral therapies for HPV infection are still limited. As a result, HPV infection continues to pose a substantial burden on public health.

To date, more than 200 HPV genotypes have been identified, and based on their oncogenic potential, they are categorized into high-risk HPV (HR-HPV) and low-risk HPV (LR-HPV) types. For the majority of HPV-infected individuals, the infection is cleared by the immune system within 1 or 2 years after initial infection (Ho et al., 1998). However, a subset of infected individuals experience persistent or latent infection lasting for years, which contributes to malignant transformation. It is now well established that persistent HPV infection plays a critical role in the development of cervical, anogenital, and head and neck cancers (Gillison et al., 2000; Radley, Saah & Stanley, 2016). Multiple studies have suggested that the repeated detection of the same carcinogenic HPV type over time may be particularly critical for cervical carcinogenesis (Schiffman et al., 2005; Wallin et al., 1999).

There are regional and population-specific differences in the incidence, persistence and clearance rates of HPV infection (Koshiol et al., 2008; Li et al., 2021; Soto-De Leon et al., 2014; Zhou et al., 2022). HPV vaccination, when administered to prepubertal girls and boys, has proven to be highly effective in preventing HPV infection and related diseases (Paavonen et al., 2009). Furthermore, China successively introduced the bivalent (2v), quadrivalent (4v), and nine-valent (9v) HPV vaccines between July 2017 and the end of 2018 (Yin, 2017). Therefore, a reevaluation of the local incidence, persistence, and clearance rates of HPV infections is needed to better understand the impact of HPV vaccination and provide a scientific basis for local immunization strategies. The study aimed to investigate the incidence, persistence, and clearance rates of HPV infection among gynecologic outpatients in Yunnan province between October 2019 and August 2023. The primary goal was to generate epidemiological data to identify HPV subtypes that are more likely to persist and those that are readily cleared in the local population.

Materials and Methods

Study population

Data were collected from women who visited the gynecological clinic of the First People’s Hospital of Yunnan Province between October 2019 and August 2023 and were advised to undergo routine HPV testing. Women were eligible if they met the following criteria: (i) attended the hospital’s gynecological clinic; (ii) provided cervical brush samples for HPV DNA testing; and (iii) were aged 15 years or older. Patients were excluded if they met any of the following criteria: (i) male sex; (ii) missing relevant clinical data; and (iii) age younger than 15 years. This study was approved by the ethics committee of the First People’s Hospital of Yunnan Province (No. KHLL2023-KY068). All study procedures complied with the Declaration of Helsinki, and written informed consent was obtained from all participants prior to study initiation, in accordance with the requirements of the research ethics committee. In this study, the terms 2vHPV, 4vHPV, and 9vHPV refer to HPV genotype groups covered by the bivalent, quadrivalent, and nonavalent HPV vaccines, respectively (as licensed in China). These terms are used solely for categorizing and analyzing the prevalence and distribution of HPV types according to their vaccine coverage, rather than to indicate participants’ vaccination status or to classify patients into groups. The following variables were collected: patient ID, date of HPV testing, age at HPV testing, and HPV type results. All data were de-identified prior to analysis. Details are presented in Fig. 1. Portions of this text were previously published as part of a preprint (https://www.researchsquare.com/article/rs-4606669/v1).

Figure 1 Flow chart showing the inclusion and deletion of participants.

Cervical sample collection

Vaginal administration and irrigation were prohibited for three days prior to sampling, and vaginal intercourse was prohibited for twenty-four hours. Using a cervical brush, exfoliated cervical cells were collected and stored at 4 °C in sterile sample tubes containing cell preservation solution (HEALTH Gene Technologies Co., Ltd., Ningbo, China).

DNA extraction

DNA was extracted from the sample within 24 h of collection using a nucleic acid extraction kit (HEALTH Gene Technologies Co., Ltd., Ningbo, China) according to the manufacturer’s instructions. Samples containing cervical exfoliated cells in preservation solution were vortex-mixed, transferred to a 1.5 mL sterile tube, and centrifuged at 12,000 g for 5 min. The supernatant was discarded, and the cell pellet was washed with 500 µL of PBS to remove residual contaminants such as blood or mucus. Finally, the pellet was lysed for 15 min at 100 °C in 200 µL of 5% Chelex-100 chelating resin, followed by centrifugation at 12,000 g for 5 min.

HPV testing and genotyping

HPV genotyping was performed using an HPV Genotyping Kit (HEALTH Gene Technologies Co., Ltd., Ningbo, China) according to the manufacturer’s instructions. Specific primers targeting the human β-globin locus, plasmid pcDNA 3.1(+) and the early HPV genes E6, E7, and E1 were designed. Polymerase chain reaction (PCR) amplification was performed in a Veriti heat cycler (Applied Biosystems, Foster City, CA, USA) using the following program: 55 min at 42 °C, pre-denaturation for 8 min at 94 °C, 35 cycles of 94 °C for 30 s, 60 °C for 30 s, and 70 °C for 1 min, followed by a final 1-minute extension at 70 °C. The reaction mixture contained DNA template (9 µL), PCR Master Mix (9 µL), and Taq DNA polymerase (2 µL). PCR amplicon was subsequently analyzed using a 3500DX Genetic Analyzer (Applied Biosystems, Foster City, CA, USA), which identifies 25 HPV subtypes by capillary electrophoresis based on product length in a single run. These subtypes include 15 HR-HPVs (16, 18, 31, 33, 35, 39, 45, 51, 52, 56, 58, 59, 68, 73, and 82), three probable HR-HPVs (26, 53, and 66), and seven LR-HPVs (6, 11, 42, 43, 44, 81, and 83). To exclude false-negative results due to insufficient sample, amplification of pcDNA and human β-globin was performed as internal controls to monitor the PCR process.

Statistical analysis

Participants who underwent HPV retesting within 24 months of their initial test were included to compute the incidence, persistence, and clearance rates. The 24-months follow-up was divided into four intervals: 0–6, 6–12, 12–18, and 18–24 months after the initial test.

For incident infection, the denominator was the number of individuals who tested negative for HPV at baseline and had a follow-up test within two years, and the numerator was the number who tested positive for any HPV subtype at the second test.

Persistence rates were calculated as the number of individuals who tested positive for the same HPV subtype at both baseline and the follow-up test (numerator) divided by the total number who were positive at baseline and retested within two years (denominator).

Clearance rates were calculated as the number of individuals who tested negative for the same HPV type at follow-up (numerator) divided by the number who tested positive for that type at baseline and were retested within 24 months (denominator). The statistical software SPSS, version 26.0 (SPSS, Inc., Chicago, IL, USA), was used for all statistical analyses, and a P < 0.05 was considered statistically significant.

Results

Overall characteristics of women with multiple HPV results among gynecological outpatients

A total of 45,149 individuals who met the inclusion criteria were analyzed to examine the prevalence of HPV infection in the region. Of these, 10.01% (n = 4,521) underwent two or more HPV tests within a 24-month period (Fig. 1). Among those who underwent multiple HPV tests, 76.97% (n = 3,272) had exactly two tests, with the number of tests per woman ranging from one to eight. The mean age was 40.09 ± 10.30 years. Participants were stratified into the following age categories: <30, 30–39, 40–49, 50–59, and ≥60 years, which accounted for 15.86%, 34.97%, 30.55%, 15.00%, and 3.63% of the cohort, respectively. Among those who underwent HPV retesting, 39.50% had a single infection and 16.55% had multiple infections (Table 1). Among individuals who retested positive for HPV, 42.00% tested positive for HR-HPV, and 14.05% for LR-HPV (Table 1).

Overall characteristics of HPV incidence

Among women who underwent repeat HPV testing within a 24-month period, the overall incidence rate of any HPV, 2vHPV, 4vHPV, and 9vHPV were 36.84% (95% confidential interval (CI) [34.72%–38.96%]), 4.43% (95% CI [3.52%–5.33%]), 5.64% (95% CI [4.62%–6.65%]), and 18.42% (95% CI [16.71%–20.13%]), respectively (Table 2). The majority of incident infections occurred in the 30–39 and 40–49 age groups. The five most frequently detected incident HPV subtypes were HPV 52, 51, 81, 58, and 16 (Fig. 2A). Among HR-HPV, the subtypes with the highest incidence rates were HPV 52 (5.59%), HPV 51 (3.72%), and HPV 58 (3.37%). Among LR-HPVs, HPV 81 (3.47%), HPV 43 (2.67%), and HPV 42 (2.52%) were the most prevalent (Fig. 2A). The incidence rates of any HPV infection were 13.14%, 15.60%, 8.76%, and 11.83% among women retested at 0–6, 6–12, 12–18, and 18–24 months, respectively (Fig. 3). HPV 52, 51, 58, and 53 were the most frequently detected subtypes during the 0–6 and 6–12 month intervals. Longer screening intervals were associated with higher incidence rates of HPV 16 and 39. Across all follow-up intervals, the most prevalent LR-HPV subtypes were HPV 81, 42, 43, and 44. A progressive increase in the incidence of HPV 42 was observed with longer retesting intervals, suggesting that HPV 42 may require more time for natural clearance.

Overall characteristics of HPV persistence

Among women who underwent repeat HPV testing within a 24-month period, the overall persistent rates of any HPV, 2vHPV, 4vHPV, and 9vHPV were 55.56% (95% CI [53.62%–57.50%]), 7.10% (95% CI [6.10%–8.10%]), 7.93% (95% CI [6.88%–8.98%]), and 28.57% (95% CI [26.81%–30.33%]), respectively (Table 3). The persistence rates in the <30, 30–39, 40–49, 50–59, and ≥60 years age groups were 56.60%, 49.94%, 49.43%, 69.23%, and 78.99%, respectively, demonstrating a pronounced age-related increase in persistent HPV infection. The five most frequently persistent HPV subtypes were HPV 42, 52, 58, 81, and 56 (Fig. 1B). Among HR-HPVs, the subtypes with the highest persistence rates were HPV 52 (47.83%), 58 (46.88%), and 56 (41.18%). Among LR-HPVs, HPV 42 (51.82%), 81 (45.65%), and 44 (32.93%) had the highest persistence rates (Fig. 1B). The persistent rates of any type of HPV were 12.57%, 13.26%, 6.62%, and 6.37% among women retested at 0–6, 6–12, 12–18, and 18–24 months, respectively (Fig. 4). Across the different time intervals, persistent infections were more prolonged for HPV 58 and 52 among HR-HPVs, and for HPV 42 and 81 among LR-HPVs. Among patients who tested positive for HPV at the first screening and were found to carry different HPV genotypes at the second screening (233, 9.19%), dynamic changes in genotype profiles were frequently observed (Table S1). Overall, 136 (58.37%) showed co-infection with new genotypes while retaining at least one of their baseline types, whereas 97 (41.63%) experienced complete genotype replacement, with all baseline types cleared and only new genotypes detected at follow-up. Within the co-infection group, 90 (38.63%) transitioned from single-type infection at baseline to multiple-type infection at follow-up. HR-HPV accounted for the majority of newly detected genotypes, particularly genotypes HPV 52 and 58 (Fig. S1).

Table 1 Prevalence of HPV infection among women with multiple HPV results.

Characteristic	<30	30–39	40–49	50–59	≥60	Total (n)	%	
HPV negative	247	754	679	262	45	1,987	43.95%	
HPV positive	470	827	702	416	119	2,534	56.05%	
Single infection	294	589	522	301	80	1,786	39.50%	
Multiple infection	176	238	180	115	39	748	16.55%	
HR-HPV	312	635	536	320	96	1,899	42.00%	
LR-HPV	158	192	166	96	23	635	14.05%	
Total	717	1,581	1,381	678	164	4,521	100.00%	

Table 2 Type-specific HPV incidence among women who retested for the virus.

HPV subtypes	<30	30–39	40–49	50–59	≥60	Total	Incidence rates	95% CI	
Any HPV	119	235	202	138	38	732	36.84%	34.72%–38.96%	
2vHPV	13	28	23	17	7	88	4.43%	3.52%–5.33%	
4vHPV	17	35	32	21	7	112	5.64%	4.62%–6.65%	
9vHPV	54	102	114	79	17	366	18.42%	16.71%–20.13%	

Figure 2 Overall HPV genotype distribution of (A) HPV incidence, (B) persistence, and (C) clearance.

Figure 3 Overall HPV genotype distribution of HPV incidence within 6 months (A), 6–12 months (B), 12–18 months (C), and 12–24 months (D).

Table 3 Type-specific HPV persistence among women who retested for the virus.

HPV subtypes	<30	30–39	40–49	50–59	≥60	Total	Persistent rates	95% CI	
Any HPV	266	413	347	288	94	1,408	55.56%	53.62%–57.50%	
2vHPV	35	67	41	27	10	180	7.10%	6.10%–8.10%	
4vHPV	42	72	42	34	11	201	7.93%	6.88%–8.98%	
9vHPV	149	217	178	138	42	724	28.57%	26.81%–30.33%	

Figure 4 Overall HPV genotype distribution of HPV persistence within 6 months (A), 6–12 months (B), 12–18 months (C), and 12–24 months (D).

Overall characteristics of HPV clearance

Among women who underwent repeat HPV testing within a 24-month period, the overall clearance rates for any HPV, 2vHPV, 4vHPV, and 9vHPV were 74.43% (95% CI [72.73%–76.13%]), 13.46% (95% CI [12.13%–14.79]), 17.68% (95% CI [16.19%–19.17%]), and 44.44% (95% CI [42.50%–46.37%]), respectively (Table 4). The clearance rates among HPV-positive individuals in the <30, 30–39, 40–49, 50–59, and ≥60 age groups were 77.87%, 76.66%, 75.64%, 67.55%, and 62.18%, respectively. HPV clearance rates showed a progressive decline with increasing age. The five HPV subtypes most likely to be cleared were HPV 26, 83, 11, 82, and 44 (Fig. 1B). Among HR-HPVs, the subtypes with the lowest clearance rates were HPV 58 (53.66%), 52 (53.99%), and 35 (62.50%), while for LR-HPVs, the lowest clearance rates were observed for HPV 42 (56.93%), 81 (62.50%), and 43 (85.15%). The majority of HPV infections were cleared rapidly, with approximately two-thirds resolving within the first year (Fig. 5).

Table 4 Type-specific HPV clearance among women who retested for the virus.

HPV subtypes	<30	30–39	40–49	50–59	≥60	Total	Clearance rates	95% CI	
Any HPV	366	634	531	281	74	1,886	74.43%	72.73%–76.13%	
2vHPV	73	103	89	56	20	341	13.46%	12.13%–14.79	
4vHPV	120	134	108	63	23	448	17.68%	16.19%–19.17%	
9vHPV	255	382	291	146	52	1,126	44.44%	42.50%–46.37%	

Figure 5 Overall HPV genotype distribution of HPV clearance within 6 months, 6–12 months, 12–18 months, and 12–24 months.

Discussion

The importance of persistent HPV infection in the carcinogenesis of cervical cancer is now widely acknowledged. Cervical cancer remains one of the most common malignancies among women, despite substantial efforts through HPV vaccination programs. Moreover, there remains a pressing need to develop efficient therapies for both HPV infections and HPV-related diseases.

In the present study, the overall incidence rates of any HPV, 2vHPV, 4vHPV, and 9vHPV were 36.84%, 4.43%, 5.64%, and 18.42%, respectively. The 30–39 and 40–49 years age groups had the highest incidence rates. HPV 52 had the highest incidence rates, followed by HPV 51, 81, 58, and 16. To the best of our knowledge, this represents the most recent report on the incidence of HPV infections in Yunnan. In comparison, the HPV subtypes with the highest incidence rates in Yunnan between 2012 and 2014 were HPV 16, 52, and 83 (Zou et al., 2016). This suggests that, over time and with the introduction of HPV vaccination, the most common local HPV subtypes are gradually shifting. The overall incidence rate in our study is higher than that reported among women in Guangdong (10.58%) and elderly women in Sweden (2.4%) (Lanner & Lindstrom, 2020). In a another survey, young Dutch women aged 16 and 29 had an overall incidence rate of 45% (Mollers et al., 2013). Differences in the follow-up duration, study populations, time frames, and other factors may account for the observed variation in incidence rates.

The overall persistence rates of any HPV, 2vHPV, 4vHPV, and 9vHPV in our study were 55.56%, 7.10%,7.93%, and 28.57%, respectively. The highest persistence rates were observed in the <30 and ≥60 year age groups. HPV 42, 52, 58, 81, and 56 exhibited the highest persistence rates, with LR-HPV 42 and 81 and HR-HPV 58 and 52 being particularly persistent compared to other subtypes. In addition, the dominant persistent HPV subtypes have shifted compared with data from 2012–2014. Between 2012 and 2014, the subtypes with the highest persistence rates were HPV 43 (16.7%), 16 (15.3%), 58 (14.0%), 6 (12.2%), and 52 (8.5%) (Zou et al., 2016). It has also been reported that the median HPV persistence rates at 12 and 24 months following therapy were 13% and 4%, respectively (Hoffman et al., 2017). The HPV persistence rate in our study was higher than those reported in Daqing (34.12%) (Li et al., 2017), Heilongjiang (33.9%) (Liu et al., 2020), Wuhan (46.51%) (Pan et al., 2022), Mexico (38.00%) (Oyervides-Munoz et al., 2020), and the Netherlands (44.10%) (Van der Weele et al., 2016). In Guangdong, the persistence rate was 47.55% at 24 months, with HPV 42, 53, and 81 being the most common persistent subtypes (Li et al., 2021). HPV 42 was more frequently persistent than other subtypes, consistent with our findings (Li et al., 2021). Although not included in current vaccine, HPV 42 has been found to be significantly associated with cervical intraepithelial neoplasia (Regauer, Reich & Kashofer, 2019) and digital papillary adenocarcinoma (DPA) (Leiendecker et al., 2023), suggesting that future immunization programs should consider this subtype. In addition to HPV genotype and host-related factors, co-infections with other viruses such as human immunodeficiency virus (HIV), hepatitis B virus (HBV), and hepatitis C virus (HCV) may also influence the persistence of HPV infection and the progression toward cervical neoplasia (Pavone et al., 2024). Previous studies have highlighted that people living with HIV exhibit a markedly high prevalence of anal HPV infection (86.4%) and HPV-associated squamous intraepithelial lesions, underscoring the synergistic effect of multiple viral infections in promoting neoplastic transformation (Sambo et al., 2025). Although our study did not collect data on these co-infections, their potential impact should be considered when interpreting the persistence and clearance rates observed in our population.

The overall clearance rates of any HPV, 2vHPV, 4vHPV, and 9vHPV in our study were 74.43%, 13.46%,17.68%, and 44.44%, respectively. Clearance rates increased with younger age, showing a progressive decline in older individuals. The HPV subtypes most readily cleared were 26, 83, 11, 82, and 44. Among HR-HPVs, the subtypes least likely to be cleared were 58, 52, and 35, while among LR-HPVs, HPV 42, 81, and 43 were the least likely to be cleared. Among Colombian women, HPV 18 and HPV 31 subtypes were reported to have the lowest clearance rates (Soto-De Leon et al., 2014). The 24-month HPV clearance rate among women were 40.4% in Ouro Preto of Brazil (Miranda et al., 2013) and 52.44% in Guangdong (Li et al., 2021). Among rural Uyghur women in China, the HPV clearance rate was 59.74% at 12 months and 69.13% at 24 months (Tuerxun, Abudurexiti & Abulizi, 2023). Differences in HPV subtype, viral load, geographic location, and other factors may contribute to these variations. Apart from HPV 16 and 18, HPV 58 was the most commonly detected subtype in individuals with high-grade squamous intraepithelial lesion (HSIL) and was a critical risk factor for cervical disease progression (Seong et al., 2021; Song et al., 2013; Yi et al., 2022). Given that HPV58 was the most persistent and challenging to eradicate in Yunnan, it should be prioritized for monitoring and prevention, particularly in older populations. Our findings are limited by several factors. First, the patient sample size was relatively small, and larger cohorts will be required in future studies to confirm our findings. Since Yunnan is a multiethnic province, more accurate information could be obtained through studies involving multiple centers and diverse ethnic groups. Second, the study focused exclusively on gynecologic outpatients, which may have contributed to the higher proportion of persistent infections observed. Third, we were unable to account for additional variables associated with HPV incidence, persistence, and clearance, such as participants’ geographic location,, sociodemographic and behavioral factors, treatment history, and cervical cytology or histology results.

Conclusions

In conclusion, women in Yunnan demonstrated a higher likelihood of acquiring and maintaining infections, and a lower likelihood of clearing infections with HPV 58, 52, 42, and 81. Older adults were more prone to develop persistent HPV infections, whereas younger individuals were more likely to achieve viral clearance.

Supplemental Information

Supplemental Information 1 STROBE checklist

Supplemental Information 2 Raw data

Supplemental Information 3 Codebook

Supplemental Information 4 Follow-up outcomes for baseline HPV-positive patients (n, %) and proportion involving high-risk HPV genotypes (n, %)

Supplemental Information 5 Newly detected HPV genotypes at second screening

The stacked bars show the number of cases for each genotype, divided into co-infection (new genotypes added while baseline types were retained) and complete genotype replacement (baseline types cleared, only new genotypes detected).

Additional Information and Declarations

Competing Interests

Author Contributions

Human Ethics

Data Availability

The authors declare there are no competing interests.

Yafei Huang conceived and designed the experiments, performed the experiments, analyzed the data, prepared figures and/or tables, and approved the final draft.

Xiangcong Wei analyzed the data, prepared figures and/or tables, and approved the final draft.

Yan Guo performed the experiments, prepared figures and/or tables, and approved the final draft.

Ting Su performed the experiments, prepared figures and/or tables, and approved the final draft.

Qiuting Duan performed the experiments, prepared figures and/or tables, and approved the final draft.

Xin Fan performed the experiments, authored or reviewed drafts of the article, and approved the final draft.

Jinxiu Wan analyzed the data, authored or reviewed drafts of the article, and approved the final draft.

Yufan Zhang analyzed the data, authored or reviewed drafts of the article, and approved the final draft.

Guiqian Zhang performed the experiments, authored or reviewed drafts of the article, and approved the final draft.

Yi Sun analyzed the data, authored or reviewed drafts of the article, and approved the final draft.

Ya Xu conceived and designed the experiments, authored or reviewed drafts of the article, and approved the final draft.

The following information was supplied relating to ethical approvals (i.e., approving body and any reference numbers):

This research was approved by the ethics committee of the First People’s Hospital of Yunnan Province (No. KHLL2023-KY068).

The following information was supplied regarding data availability:

The raw data are available in the Supplemental Files.

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
