# Peer review of "Incidence, persistence, and clearance of cervical human papillomavirus infection among gynecological outpatients in Kunming, Yunnan, China, 2019–2023: a retrospective cohort study"

_PeerJ, doi:10.7717/peerj.20215_

## Round 0.1 · original submission · Major Revisions

**Language Note:** The review process has identified that the English language must be improved. PeerJ can provide language editing services - please contact us at [email protected] for pricing (be sure to provide your manuscript number and title). Alternatively, you should make your own arrangements to improve the language quality and provide details in your response letter. – PeerJ Staff

Reviewer 1 ·

Basic reporting

The article investigates the incidence, persistence, and clearance rates of cervical human papillomavirus (HPV) infections in gynecological outpatients. The study highlights the high incidence of transient HPV infections and the importance of regular screening, as persistent infections are more closely linked to cervical cancer risk. The findings contribute valuable insights into the natural history of HPV and support public health efforts in HPV vaccination and cervical cancer prevention strategies.
In my opinion, the structure of the article is appropriate, and the tables and figures are clear. However, I believe it should be edited by a native speaker, as it contains minor language errors that make it harder to read.

Experimental design

Research questions are well-defined, relevant and meaningful.

The authors assumed that in HPV-infected patients, the infection either persists or clears. However, in the analyzed groups, it is also possible that infection occurred after the first testing. I understand that the authors likely have such data, so why was this not presented in the study? For example, what percentage of patients initially infected with one HPV subtype showed co-infection with additional subtypes during the second testing?
I believe that such information would enrich the manuscript and allow for more comprehensive conclusions.

Another comment concerns the abbreviations HPV, 2vHPV, 4vHPV, and 9vHPV – at times they refer to vaccines, and at other times to patient groups. This needs to be clarified in the Materials and Methods section.

Validity of the findings

no comment

·

Basic reporting

Our colleagues have produced a paper whose aim is to study the presence of HPV viruses in female genitalia by age groups, but they have also achieved other important results such as verifying that clearance in young people is more likely than in the elderly population. They have also verified that multiple infections often occur. The abstract is a good summary of the paper. The next section provides a general overview of papilloma virus infection, also correctly framing the most oncogenic strains. It is suggested to add whether other infections such as HIV, HCV, HBV also co-occur (doi.org/10.3390/diagnostics15020198 to be read and cited in the bibliography). In fact, an infection by multiple viruses can more easily lead to the onset of squamous cell neoplasia. Materials and methods written with extreme clarity and perfectly replicable in any hospital. The results and the bibliography have generated discussion. It is not surprising that the age group from 29 to 59 is the most at risk given that it is the age of the greatest number of sexual intercourses, although we must say that there are people who through contaminated underwear have developed a contamination that we often find perianal and perineal without anal or vaginal contamination. After, however, a detection of HPV contamination is essential follow-up to avoid squamous cell carcinoma and we absolutely agree on vaccination even if it only covers nine strains of the virus. Excellent English, good bibliography, good iconography

Experimental design

Colleagues have conducted a research that is absolutely reproducible for the meticulousness with which it has been described on an entire province with a large part of the population that has given important results.

Validity of the findings

The results of the research are absolutely shareable and can lead to an education of both the male and female population to use protection systems to avoid contamination by HPV and not only

Additional comments

no comments

---

## Round 0.2 · accepted · Accept

Thank you for revising your manuscript to address the reviewers' concerns. Reviewer 2 now recommends acceptance and I am satisfied with your response to the earlier comments of reviewer 1. The manuscript is now ready for publication.

·

Basic reporting

The initial text has been modified according to reviewers' suggestions, but the original structure has not been distorted. It's an excellent read. Endorsement for publication.

Experimental design

I don't see critical issues

Validity of the findings

The results described in the conclusions are highly significant.

Additional comments

no comments, congrats